# Femtosecond Laser Surface Cleaning for Diamond Segmented Drill Bit Manufacturing

Attila Zsolt Kenéz [1,2,*], Éva Lublóy [3], Gyula Bagyinszki [4] and Tamás Földes [5]

1    Doctoral School on Materials Sciences and Technologies, Óbuda University, H 1034 Budapest, Hungary
2    Hilti Tool Ltd., H 6000 Kecskemét, Hungary
3    Department of Structural Mechanics, Faculty of Civil Engineering, Budapest University of Technology and Economics, H 1521 Budapest, Hungary
4    Bánki Donát Faculty of Mechanical and Safety Engineering, Department of Materials Technology, Óbuda University, H 1081 Budapest, Hungary
5    TOMOGEO Ltd., H 5000 Szolnok, Hungary
*    Correspondence: kenez.attila@uni-obuda.hu

**Abstract:** Microsecond and nanosecond lasers have been studied in the past for laser cleaning applications and, today, femtosecond lasers are also being used successfully for removing paint, rust, and surface contamination. For diamond segmented drill bits, it may be also necessary to improve the mechanical properties of the laser-welded joint, i.e., to increase the tensile strength and toughness. Therefore, in this study, we investigated the possibility of using femtosecond lasers to clean the surface before laser welding to see what effect it has on the mechanical properties of the joint. The end surface of the thin-walled tube was pretreated to remove grease and oil before laser-beam welding a powder metallurgical segment onto it and the results are compared to an untreated sample. The laser-welded seams were investigated by micro-computer tomography, break-out test, and optical microscopy. Any defects in the seams were analyzed and, according to the results obtained in this study, no cracks were found by computer tomography, a shade of grey diagram shows, and all the pre-treated samples had a higher absorption than the untreated sample. Four of the six treating parameters had a significant effect, +30% on average, and two treating parameters had a positive effect, +13.5% on average, compared to the untreated sample. In addition, the break-out values showed that only one treating parameter had a significantly, +19%, higher effect than the other treating parameters. This test showed different results from the micro-CT scan. The optimal process parameters for oil and grease removal are discussed in the conclusion.

**Keywords:** laser surface cleaning; femtosecond laser; ablation; laser-beam welding; mechanical testing; micro-computer tomography testing

## 1. Introduction

Over the past few years, femtosecond laser surface processing has been studied to improve the surface properties of various kinds of materials. Among many research areas, most studies have focused on changing the wetting properties of surfaces [1–4]; depending on the application, femtosecond laser treatment can produce surfaces with wetting properties that vary between hydrophilic and hydrophobic [5–7]. For example, when glueing or painting, the adhesion property is important; conversely, when increasing corrosion resistance, repulsion is important; when mixing different solvents, the dissolution in each other can influence. Femtosecond laser treatment can also be used to improve tribological properties [8]. Another major area of investigation is micromachining [9–14], where nano- and microscale surface structures can be created, as femtosecond lasers are flexible and well-controlled. The nano- and microscale structures created can be used in a wide range of applications, as the change in surface composition is often negligible [15]. The femtosecond laser is also suitable for surface cleaning as an effective means of removing surface coatings,

oxides, and surface contaminants from paintings, sculptures, and components [16–18]. It is also suitable for welding alumina or glass, where either surface oxide layer cracking or local melting is required [19,20]. It is also suitable for different treatment applications, e.g., steel surface hardening or etching channels into different material surfaces [20–22]. There have been limited studies on laser-welded joining of thin-walled steel tubes and powder metallurgy segments. Our investigations are aimed at improving the mechanical properties of the laser-welded joints of diamond segmented drill bits. In our first investigation, we found gas inclusions, and microcracks in the welded seam of an untreated sample. More detailed examination of these defects by scanning electron microscopy and energy-dispersive spectroscopy suggested surface contamination. The predominant presence of carbon and oxygen on the surface is clear evidence of contamination, which comes either from manufacturing or corrosion protection. During welding (due to the high temperatures involved), these elements react with the metals present and form carbide and oxide phases in an environment of bubbles caused by carbon leaving organic matter. We therefore investigated different surface cleaning methods, the results of which have been presented in a previous article [23].

In this article, we continue our investigations using a femtosecond laser to clean the surface, using the same investigation methods as in the previous article, so that we can compare the results in the future. The main aim is to study the feasibility of using a femto-second laser for surface cleaning and to understand the effects of laser parameters on cleaning. The results suggest that femtosecond lasers have the potential for use in surface cleaning applications. In this experiment, a solid-state laser source was used to weld the pretreated tube and segment together. The welded seam was investigated by computer tomography to see the effect of surface irradiation parameters on welding failures. Furthermore, the mechanical properties were investigated with a break-out test and compared to an untreated sample.

## 2. Materials and Methods

### 2.1. Materials and Welding Process

In this study, the welded joints of an E235 cold-drawn steel tube [24] and a powder metallurgy manufactured segment were investigated (chemical compositions in Table 1). The tube had a wall thickness of $2 \pm 0.15$ mm and an outer diameter of $100 \pm 0.15$ mm. The end surface of the tube was cleaned of surface contaminant (analysis result in Table 2) using a femtosecond laser source with a varying pulse repetition rate and scanning speed. After the treatment, a non-standard $3.5 \pm 0.2$ mm thick and 24 mm long segment was welded onto the end surface of the tube, using a 4 kW Trumpf TruDisk 4002-type disc laser source (parameters in Table 3). The same heat input was used for every sample, calculated using the formula described in [25]. This heat input was the same as used at serial production of the drill bit.

$$\text{Heat input (kJ/mm)} = \text{Laser power (kJ/s)}/\text{Welding speed (mm/s)} \tag{1}$$

**Table 1.** Chemical composition of tube and powder metallurgy segment (wt %).

| Materials | C | Si | Mn | P | S | Astaloy-Mo | Graphite Powder | Zn-Stearate |
|---|---|---|---|---|---|---|---|---|
| Tube | ≤0.17 | ≤0.35 | ≤1.2 | ≤0.025 | ≤0.025 | - | - | - |
| Segment | - | - | - | - | - | 99.8 | 0.2 | 0.5 |

**Table 2.** The percentage frequency of elements calculated from the energy of the X-ray photons measured on the surface of the tube.

| Elements | C | O | Fe | Na | Al | Si | S | Total |
|---|---|---|---|---|---|---|---|---|
| At % [1] | 29.39 | 12.14 | 55.22 | 1.69 | 0.20 | 0.54 | 0.81 | 100 |
| wt % [2] | 56.64 | 17.15 | 22.88 | 1.71 | 0.17 | 0.45 | 0.59 | 100 |

[1] Atomic Percentage. [2] Weight Percentage.

**Table 3.** Laser welding parameters.

| Laser Power (kJ/s) | Welding Speed (mm/s) | Shielding Gas | Gas Flow Rate (L/min) | Focal Length (mm) | Defocus (mm) | Heat Input (kJ/mm) |
|---|---|---|---|---|---|---|
| 2.8 | 50 | Ar | 10 | 20 | +0.25 | 0.056 |

## 2.2. Preparation of Test Samples

In the experiment, we used a COHERENT Monaco 1035-80-60-type laser source (60 W, 1.035 μm wavelength, maximum energy 80 μJ (at 750 kHz), a maximum pulse width 350 fs, repetition rates up to 50 MHz and beam diameter at output 2.7 ± 0.3 mm). The process setup is visible in Figure 1 and the treating process parameters are available in Table 4. The parameters were defined at the minimum and maximum repetition rate and at 750 kHz, where the laser beam source has its maximum energy [26,27]. The scanning directions of the tube were performed from four directions, 0°, 45°, 90°, and 135°, right after each other, to ensure a homogeneous surface, with the tube standing still and the laser beam following the scanning directions. Ar shielding gas was used directly during the experiment. The identifier of the sample comes from an abbreviation of femtosecond laser (FSL). The energy input was calculated using the Formulas (2), (3) and (4) and is summarized in Table 5.

(**a**) Experimental setup　　　(**b**) Snapshot during the treatment

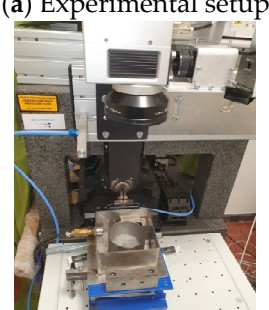

**Figure 1.** Femtosecond laser process.

**Table 4.** Process parameters.

| Samples Code | Cleaning Method | Repetition Rate | Scanning Speed (m/s) | Pulse Width (fs) | Laser Power (W) | Shielding Gas | Gas Flow Rate (L/min) | Defocus (mm) |
|---|---|---|---|---|---|---|---|---|
| FSL1 | fs | 188 kHz | 5 | 277 | 60 | Ar | 10 | 0 |
| FSL2 | fs | 188 kHz | 2.5 | 277 | 60 | Ar | 10 | 0 |
| FSL3 | fs | 750 kHz | 5 | 277 | 60 | Ar | 10 | 0 |
| FSL4 | fs | 750 kHz | 2.5 | 277 | 60 | Ar | 10 | 0 |
| FSL5 | fs | 50 MHz | 5 | 277 | 60 | Ar | 10 | 0 |
| FSL6 | fs | 50 MHz | 2.5 | 277 | 60 | Ar | 10 | 0 |

**Table 5.** Summary of energetic characteristics.

| Character | FSL1 and FSL2 | FSL3 and FSL4 | FSL5 and FSL6 |
|---|---|---|---|
| Repetition rate (Hz) | $188 \times 10^3$ | $750 \times 10^3$ | $50 \times 10^6$ |
| Peak power (W) | $1.152 \times 10^9$ | $2.888 \times 10^8$ | $4.332 \times 10^6$ |
| Energy per pulse (J) | $4 \times 20 \times 10^{-6}$ | $1 \times 80 \times 10^{-6}$ | $1 \times 1.2 \times 10^{-6}$ |
| Peak power density (W/mm$^2$) | $4.024 \times 10^8$ | $1.009 \times 10^8$ | $1.513 \times 10^8$ |

Calculation of energetic characteristics based on process parameters:

$$\text{Peak power (W)} = \text{Average power (W)}/(\text{Repetition rate (Hz)} \times \text{Pulse width (s)}), \quad (2)$$

$$\text{Energy per pulse (J)} = \text{Peak power (W)} \times \text{Pulse width (s)}, \quad (3)$$

$$\text{Peak power density (W/mm}^2) = 2 \times (\text{Average power (W)}/(\text{Repetition rate (Hz)} \times \text{Pulse width (s)} \times \text{Beam area (mm}^2))). \quad (4)$$

After fs laser treatment, the surface roughness on the tube end surface was measured using a Mitutoyo Surftest SJ-210 portable measuring instrument, with parameters of measuring range 10 mm, measuring speed 0.5 mm/s, cut-off length 2.5 mm, and the ISO 4287:1997 standard was used for the test [28]. After measuring the surface roughness, the same surface was examined using a Leica M205A stereomicroscope equipped with a Leica DMC 4500 colour camera. Then, after laser welding of the samples, the seam image was examined visually to determine whether a deviation or defect was visible on the seam surface. Then, every sample was prepared for micro-computer tomography using a WERTH Tomoscope HV 500 machine with a measurement data of 190 kV/0.13 mA, pixel spacing of 0.0646 mm, 1672 rows and 1681 columns [29]. After the images are taken, a computer algorithm reconstructs the virtual sections of the sample. Then, the microstructure was examined with a Keyence VHX-7000 digital microscope on samples polished with a 0.1 μm baize, end etched with 2% Nital. Detailed analysis was performed on front, middle, and end of the seams of the embedded samples. A Leica DMi8 inverse microscope with a Leica DMC 4500 colour camera was used to take images of microstructure with a 5× objective. In addition, a panorama image was taken to see the whole cross-section. LASX software can control the microscope and take measurements of the captured image. It measured the seam length and the size of the pores at every seam.

The break-out test was carried out at a crosshead speed of 6 mm/min using a 100 kN INSTRON 4482 electromechanical, universal material testing machine. Owing to the character of the technical test, the standard does not apply, and individual tests must be performed in each case. A device was designed to hold the parts containing the segments cut from the drill bits, where the lower edge of the segment was 1 mm before the device so that it rested on the surface of the tube and not on the seam (Figure 2a). The shape of the pressure head matched the shape of the segment (Figure 2b). The machine recorded the force–displacement diagram (Figure 3) and only Section 1 was considered, as this is the section of elastic deformation that is mechanically decisive.

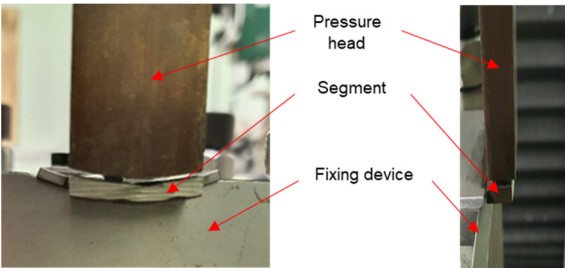

(**a**) Transversely     (**b**) In the direction of the longitudinal axis

**Figure 2.** Location of the piece to be tested in the device.

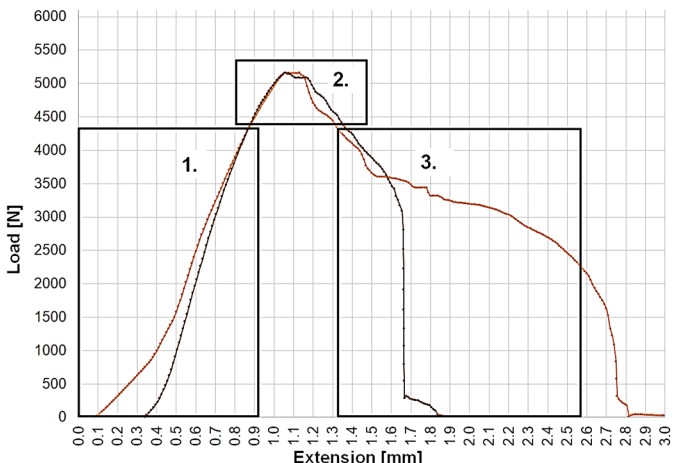

**Figure 3.** Structure of the force–displacement diagram. (1) Section of elastic deformation; (2) section of plastic deformation; (3) section of local plastic deformation where plastic deformation is limited to a narrow volume at the seam cross-section contract.

## 3. Experimental Results and Discussion

### 3.1. Macrostructure

The welded seam of six, different, cleaned and one untreated sample was visually checked (the welded seam of the untreated sample can be seen on Figure 4). No surface deviation or error that adversely affects the mechanical properties, was found on any of the samples.

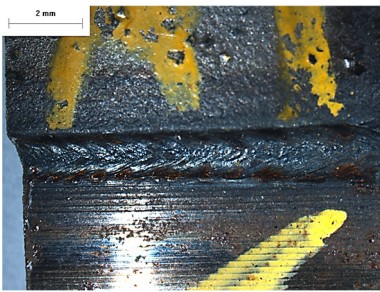

**Figure 4.** Image of the welded seam of the untreated sample.

### 3.2. Microstructure Characterization

#### 3.2.1. Surface Analysis

Surface roughness was measured on the treated surface of each sample in three measurements, 120° apart. In Figure 5, the box-plot diagram shows the results. The average surface roughness of the FSL1, FSL2, FSL5, and FSL6 samples was 15% smoother than the untreated sample, whereas that of the FSL3 and FSL4 samples was 21% rougher. In

addition, five of the six samples had a smaller standard deviation than the untreated sample. To summarize the surface roughness measurement, the FSL5 sample shows the smoothest surface, because it has the smallest mean value and the smallest standard deviation. In Figure 6, microscopic images of the treated surface are shown. Slight corrosion was observed on the surface of the UT and FSL3 samples, which was considered in the surface roughness measurement and avoided. The different treatment parameters produced different surface structures. However, all images show "horizontal lines" that may have formed during the serial production of the tube component and remained visible after laser scanning.

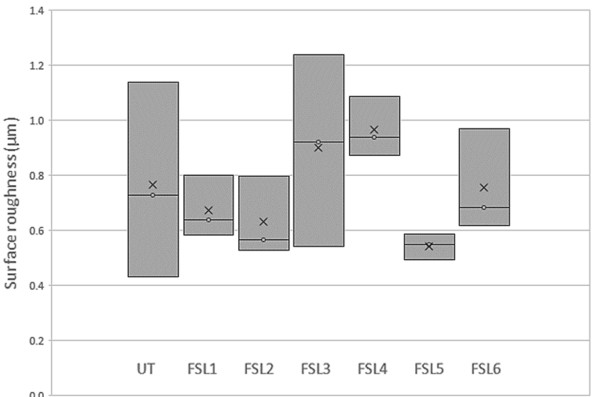

**Figure 5.** Surface roughness evaluation diagram.

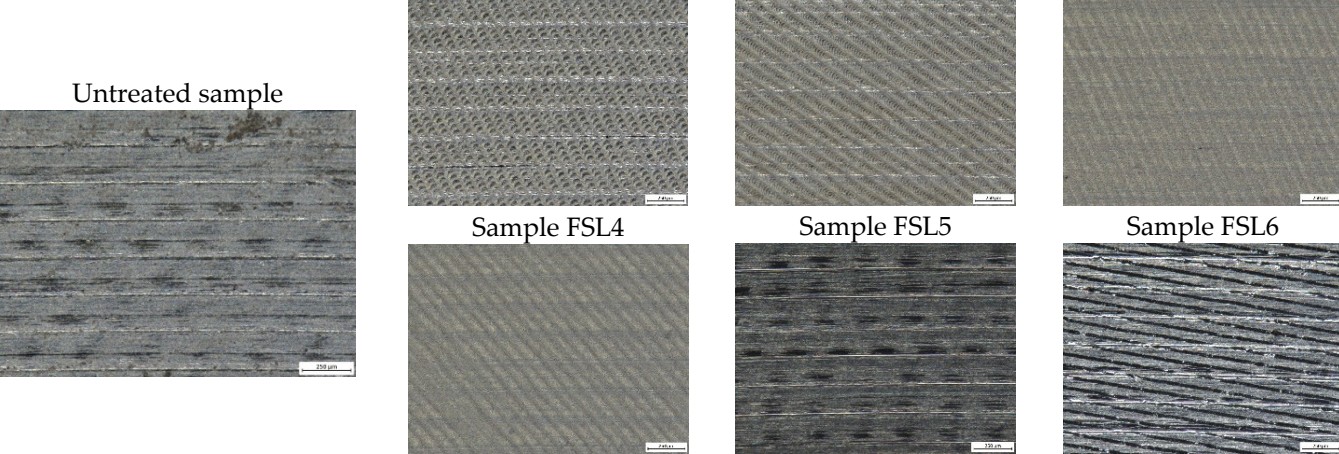

**Figure 6.** Surface morphology of different laser irradiations.

### 3.2.2. Micro-Computer Tomography

Figure 7 shows a micro-CT image of the drill bit. Seams were analyzed by examining the images of the samples, but porosities or cracks were not found. Furthermore, when the grey scale of intensity was generated, it revealed that all the pretreated samples had a higher absorption than the untreated sample. Based on the X-ray attenuation of the sutures, we used a methodology based on the statistical distribution of the grey-scale intensities to extract image features from the micro-CT images. The grey-scale intensities were directly obtained from the pixels of micro-CT images. The intensities are positive integers with values between 0 and 255, the higher the value, the higher the absorption, and they also give information on the surface's cleanliness.

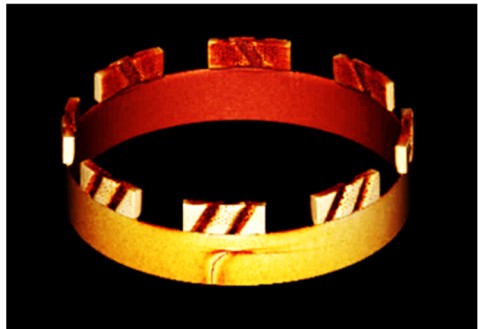

**Figure 7.** Recording of the drill bit.

Figure 8 shows the highest numbers, where the values have already been arranged in descending order starting from the highest value. It can be concluded that the lowest value is observed for UT, i.e., for untreated sample, whereas samples FSL1, FSL3, FSL4, and FSL5 had a significantly, +30% on average, higher value. Samples FSL2 and FSL6 had a positive, +13.5% on average, effect. Summarizing the micro-computer tomography, all treated samples showed better values than the untreated sample, but the FSL4 sample had highest value.

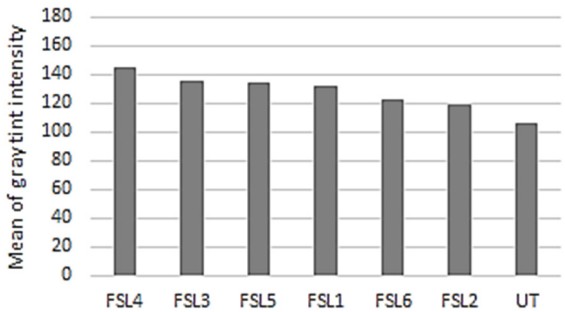

**Figure 8.** The shade of grey diagram.

### 3.2.3. Break-Out Test

Eight segments were welded onto each sample (as seen in Figure 7), and all the segments were broken. When each segment was broken, the force–displacement diagram was recorded, from which the forces and deflections at the yield point and the maximum values can be read. Figure 9 shows the force–displacement diagram for untreated and treated samples. Circles indicate the yield strength values, and triangles indicate the maximum force values. The software has shifted the curves so that the tangents drawn on the straight section near the yield point meet at the 0 mm deflection point. From a mechanical point of view, the yield strength is more important than the maximum value because, after the yield strength is exceeded, the residual deformation at the joint is formed. Data are available for both forces separately, but diagrams have been produced only for values at the yield strength. Figure 10a shows the average breaking forces; all treated samples had better values than the untreated sample. The average value of the FSL6 sample was +19% higher, with no significant difference between the other treated samples (+7.5% on average) and the untreated sample. Standard deviation (SD) and range (R) data state that sample FSL1 has the lowest SD and R-value, whereas sample FSL6 has the highest SD and R-value at breaking force (results in Table 6).

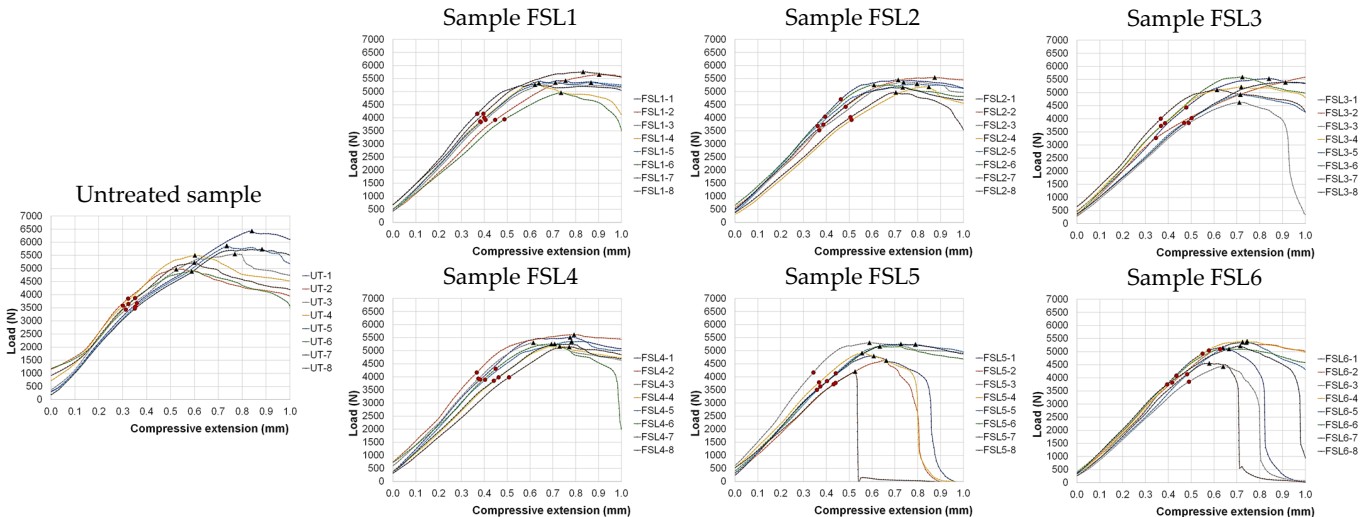

**Figure 9.** Force–compressive extension diagram per sample.

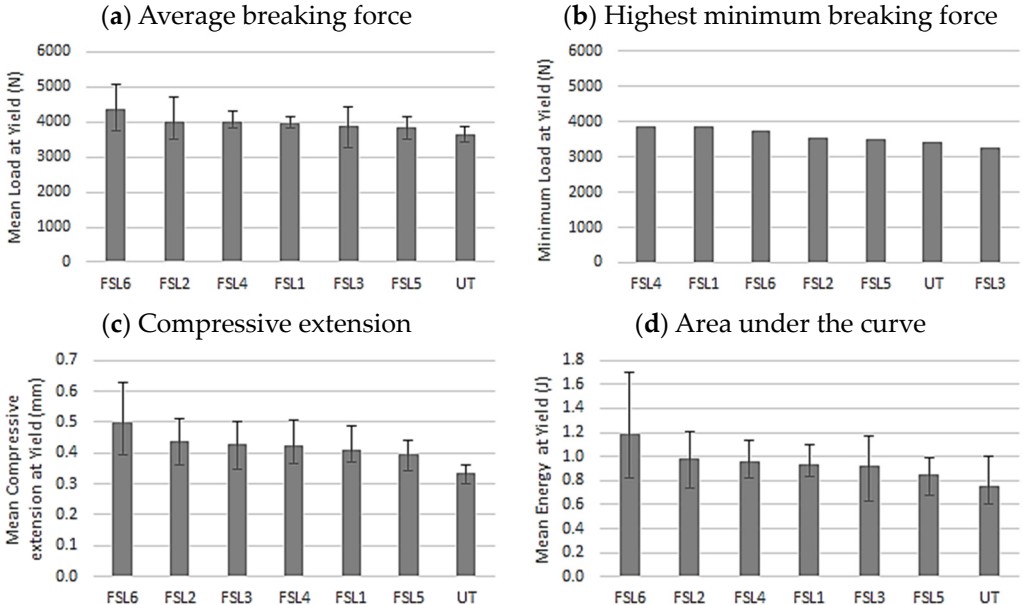

**Figure 10.** Evaluation of break-out test.

**Table 6.** Summary of break-out test (data for yield point).

| Character | | UT | FSL1 | FSL2 | FSL3 | FSL4 | FSL5 | FSL6 |
|---|---|---|---|---|---|---|---|---|
| Load (N) | SD [2] | 163 | 122 | 394 | 327 | 157 | 231 | 583 |
| Load (N) | R [3] | 444 | 317 | 1190 | 1169 | 453 | 668 | 1349 |
| C.E. (mm) [1] | SD [2] | 0.022 | 0.040 | 0.063 | 0.065 | 0.050 | 0.039 | 0.082 |
| C.E. (mm) [1] | R [3] | 0.059 | 0.119 | 0.148 | 0.155 | 0.141 | 0.098 | 0.233 |
| Energy (J) | SD [2] | 0.14 | 0.09 | 0.19 | 0.18 | 0.12 | 0.10 | 0.35 |
| Energy (J) | R [3] | 0.40 | 0.26 | 0.48 | 0.54 | 0.31 | 0.31 | 0.89 |

[1] Compressive Extension. [2] Standard deviation. [3] Range.

Figure 10c shows the compressive extensions, which show how long the joint can withstand the load. Sample FSL6 has a significantly, +49%, higher mean value, samples FSL2, FSL3, and FSL4 have +28.5%, on average, higher mean values, and samples FSL1 and FSL5 have +20%, on average, higher mean values than the untreated sample. The untreated sample has the lowest SD and R-value at compressive extension, whereas sample

FSL6 has the highest SD and R-value. Figure 10d shows the area under the curve (i.e., the fracture work), which indicates the seams' brittleness and toughness. Sample FSL6 has a significantly, +58%, higher mean value and samples FSL1, FSL2, FSL3, and FSL4 have +26.5%, on average, better results than sample FSL5, but all treated samples showed better values than the untreated sample. Sample FSL1 has the lowest SD and R-value, whereas sample FSL6 has the highest SD and R-value for area under the curve.

Figure 10b shows the minimum breaking force, which is a single value (the higher this value, the higher the strength of the seam). Samples FSL1, FSL4, and FSL6 have +10.5%, on average, better results than samples FSL2 and FSL5 (+2.5% in average). Sample FSL3 had a smaller value, −5%, than the untreated sample. The data are arranged in descending order from the highest value to show the trend and this applies to every diagram.

Summarizing the break-out test, sample FSL6 showed the highest mean value for all characteristics: mean fracture value, mean deflection, and mean area under the curve. Note that the standard deviation and range value was also the largest for this sample. In addition, samples FSL1 and FSL4 had +12% and sample FSL6 had +9% higher minimum breaking force values than the untreated sample. Based on all four characteristics, the treatment parameters of the FSL6 sample appear to be the most robust.

### 3.2.4. Optical Microscopy

After the break-out test, samples were analyzed by optical microscope. The microstructure of the welded seam corresponded to a powder metallurgical segment and steel mixed bond, i.e., mostly bainitic, and martensitic in places. Figure 11 shows images of the welded seam and the heat-affected zone boundary on the segment side, as the surface irradiation procedures only affected this area. All the samples show that the inclusions are located at the seam–segment boundary. For samples FSL2 and FSL5, it was observed that a solidification crack was visible near the boundary of the seam segment, in the middle of the seam. In sample FSL3, the crack may have resulted from the breakout test, because it started outside of the seam.

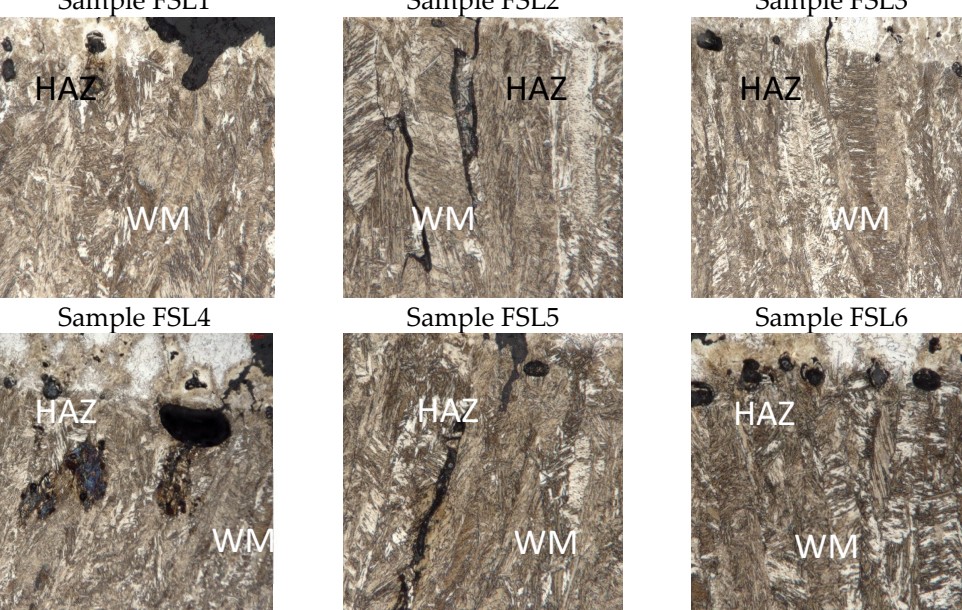

**Figure 11.** Microscopy images of the welded seams.

Figure 12 shows the pictures from the beginning of the seam, and Figure 13 shows the pictures from the end of the same seam. The images are aligned so that the welding position is shown in the same plane (indicated by the green dashed line, the bottom line in the image). The segmented side heat-affected zone, located between light and dark blue

dashed lines (middle and top lines in the pictures), is the same distance from the welding position and the same height, both at the beginning and at the end of the weld. Using the LASX software, the pores in the seam were counted. Their area was measured on three pieces for each sample, and then the ratio of the fraction of the fracture size to the length of the seam (i.e., how much of the fraction was in the seam) was determined. The results were averaged and are summarized in Table 7. It can be concluded that the surface treatments had a positive effect on the strength of the weld because the fraction of fracture in the weld was smaller in all treated samples than in the untreated sample (third line in Table 7). The average number of pores was smaller in samples FSL1, FSL3, and FSL5 than in the untreated sample. The average area of pores was smaller in samples FSL1, FSL2, and FSL3 than in the untreated sample. Summarizing the optical microscopy, sample FSL1 showed the best results in the case of all three characteristics.

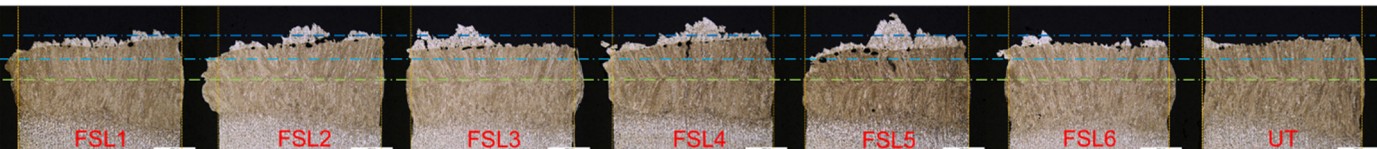

**Figure 12.** Fracture images from the front of the seam.

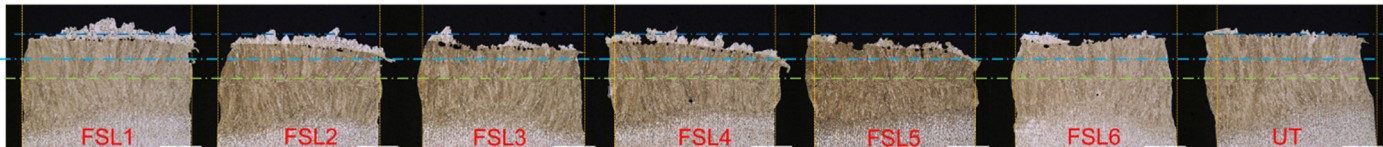

**Figure 13.** Fracture images from the end of the seam.

**Table 7.** Summary of the microscopy examination results.

| Character | UT | FSL1 | FSL2 | FSL3 | FSL4 | FSL5 | FSL6 |
|---|---|---|---|---|---|---|---|
| Average number of pores | 7.75 | 6.50 | 10.25 | 7.50 | 10.50 | 7.00 | 9.50 |
| Average area of pores (mm$^2$) | $9.987 \times 10^{-3}$ | $3.389 \times 10^{-3}$ | $8.736 \times 10^{-3}$ | $8.386 \times 10^{-3}$ | $10.016 \times 10^{-3}$ | $10.084 \times 10^{-3}$ | $10.906 \times 10^{-3}$ |
| Average of the part broken out of the seam (%) * | 36 | 0 | 6 | 14 | 14 | 8 | 19 |

* Relation to the total length of the seam.

## 4. Conclusions

This study investigated femtosecond laser irradiation to remove grease and oil from the tube surface before laser-beam welding. The results were compared to an untreated sample. The following conclusions were deduced from the analysis:

1.  The surface analysis shows one parameter (FSL5), out of six, with significantly better results than the untreated sample;
2.  Through the micro-CT analysis, the shade of grey diagram shows that four (FSL1, FSL3, FSL4, and FSL5) of the six treating parameters have a significant effect compared to the untreated sample. The highest value was shown by sample FSL4;
3.  Break-out value (BOV) is the principal qualification value in mass production, and the result of this test should be weighted;
4.  The break-out values show that only one treating parameter (FSL6) had a significant effect compared to the untreated sample;
5.  Microscopy analysis shows a single parameter (FSL1) with significantly better results than the untreated sample.

The four different test methods, surface analysis, micro-CT analysis, break-out test, and optical microscopy, showed the best results for samples irradiated with four different parameters, these being FSL1, FSL4, FSL5, and FSL6. Since break-out value is considered the most important feature, the FSL6 parameter is considered the best parameter. However, it is recommended that a larger number of samples is produced and tested with this parameter to allow a more detailed comparison.

**Author Contributions:** Conceptualization, A.Z.K.; methodology, A.Z.K.; validation, A.Z.K. and É.L.; formal analysis, É.L. and G.B.; investigation, A.Z.K. and T.F.; writing—original draft preparation, A.Z.K.; writing—review and editing, É.L. and G.B. All authors have read and agreed to the published version of the manuscript.

**Funding:** This research received no external funding.

**Data Availability Statement:** Data sharing not applicable.

**Acknowledgments:** The authors thank all colleagues who helped in the preparation, analysis, and evaluation of the samples.

**Conflicts of Interest:** A.Z.K. reports administrative support and equipment, supplies were provided by Hilti Group. The corresponding author is currently employed by Hilti Group.

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
