# Peer review of "Femtosecond Laser Surface Cleaning for Diamond Segmented Drill Bit Manufacturing"

_crystals, doi:10.3390/cryst13040672_

Round 1

Reviewer 1 Report

The authors study the laser cleaning using femtosecond laser, and 6 sets of machining parameters are tested in order to obtain the optimal parameter combination. Some critical questions are not addressed by the authors, such as the necessity of the study and the logic of choosing machining parameters.

1. The authors don’t indicate the relationship between laser processing parameters and experimental effects. 
2. Why did the authors choose the laser processing parameters in Table 1 and Table 4? What is the basis for selecting laser processing parameters? The authors did not study the regularity of laser parameters. Will it work better with other fs laser processing parameters? Some methods such Taguchi method or Response Surface Methodology could be used to design the machining parameters.
3. A more detailed discussion of the characterization of experimental results is missing. The correlation between experimental characterization values and laser cleaning and welding effects is not sufficient.
4. Does the element composition change before and after femtosecond laser cleaning? Reference or explanation should be added to the explanation part.

Below are some comments for the specific locations:
Abstract: I suggest the authors rewrite the abstract with a focus on the background, objectives, methodology, main findings, and conclusions. Please add a sentence that shows the necessity of the study. The description of the effect of the experiment in lines 23-28 is not clear. What does “24-36%/12-15%/19%” mean? It is not clear to make this statement.
Introduction: The author should introduce the research status of other types of lasers for cleaning and welding. What are the differences and advantages of fs laser cleaning and welding compared with other types of laser cleaning and welding? The author needs to provide the contributions and innovation of this study more specifically.
(75): Machining parameters FSL 1-FSL 6 in Table 1 are the same, so there is no need to use different rows to indicate that.
(88) Table 4: Laser power should be directly expressed in terms of power value.
(125) Figure 3: The figures in the picture are not clear.
(127): The pictures of the macrostructure are missing.

Reviewer 2 Report

The manuscript needs significant revision of the text, illustrations, tables and bibliography.

1.       The authors of the manuscript write that "Microsecond and nanosecond lasers have been studied in the past for laser cleaning applications", but do not provide any results of these works, nor links to these publications. In this regard, statements about the advantages of femtosecond lasers seem unfounded. It is necessary to supplement the manuscript with literature data and a critical analysis of laser cleaning applications using laser pulses of different durations, also discussing the economic feasibility of using femtosecond lasers.

2.       Three of the four co-authors of this manuscript published the article "Effect of surface cleaning on seam quality of laser beam welded mixed joints" in Case Studies in Construction Materials 18 (2023) e01904. Its topics, samples, methods and results of diagnostics are close to those discussed in the reviewed manuscript (In this study, “different surface cleaning methods were used to remove greases and oils from the surface of the E235 steel tube component of the diamond segmented drill bit before laser beam welding"). Some of the tables and figures in the article "Effect of surface cleaning on seam quality of laser beam welded mixed joints" completely coincide with those presented in the reviewed manuscript. It would be logical to link to this article and compare the results of chemical and laser cleaning.

3.       The purpose of Table 2 is not entirely clear. Laser welding parameters are absolutely identical for all six samples.

4.       In table 3, you must specify what the designation is (m / m%). If this is a mass fraction, then the correctness of converting atomic percentages into weight fractions is questionable. The main part of carbon is present on the surface of the sample in the form of hydrocarbons and, possibly, carbides. However, hydrogen itself is missing from the list of elements found at the surface of the tube.

5.       All captions under the figures are too concise, which makes it difficult to read the article.

6.       The data in the tables are often repeated, do not carry a semantic load (for example, the bottom line in table 7) or are given without taking into account the accuracy of measurements (for example, the second line in table 7).

7.       The manuscript says that “The femtosecond laser is also suitable for surface cleaning as an effective means of removing surface contaminants, oxides, and surface coatings [16-18], suitable for welding [19, 20] and different treatment applications [21, 22]. There are limited number of studies on the laser welded joint of thin-walled steel tubes and powder metallurgy segments. In our previous tests, we found gas inclusions, and microcracks." Most of these publications are not directly related to the experiments described in the manuscript under review and are not discussed further in its text. At the same time, references to their previous research and publications on surface cleaning of thin-walled steel tubes and powder metallurgy segments are not provided.

8.       The formula for calculating “Peak power density” does not take into account the Gaussian distribution of the laser beam energy

9.       It is necessary to give explanations about the two curves (black and brown), and the representation of the dependencies for the three areas 1,2 and 3 in Figure 3, taking into account the fact that areas 2 and 3 are not discussed in the text of the manuscript. Raises questions and a significant difference in the course of the curves on the Displacement - force diagram in the reviewed manuscript and the article "Effect of surface cleaning on seam quality of laser beam welded mixed joints" by the same authors.

10.    The text of the manuscript says that "According to Table 6, the Untreated sample has the lowest SD and R-value", but this does not quite agree with the numerical data given in Table 6.

11.    It is difficult to recognize the conclusions about the optimal modes of laser surface treatment as justified, since all the differences in the parameters in Figure 9 are within the accuracy of their determination.

12.    The information in the bibliography needs to be improved. For example, reference 22 - indicate volume and pages, references 24 and 26 - indicate the names of journals (Journal of Manufacturing Processes and Radiotherapy and Oncology), volume and page numbers.

1The general impression of reading the manuscript. The text is written rather chaotically. The material itself is of limited interest and does not contain a critical analysis of the results obtained. Conclusions about the optimal modes of surface treatment are not clear. I would like to agree with the authors of the manuscript that "However, it is recommended that larger numbers of samples are produced and tested with this parameter to allow a more detailed comparison".

Round 2

Reviewer 1 Report

The revision shows improvement of the manuscript.

Table 4 contains the minimum (188 kHz), maximum (50 MHz) of the repetition rate. At 750 kHz there is a maximum energy of the laser source (rows 99-100).

The selection basis of femtosecond laser cleaning parameters is not clearly explained. Looking at the data in Table 5, it can not reflect the maximum energy of the laser source when the pulse repetition frequency is 750KHz. In addition, the influence of laser cleaning parameters on the cleaning effect should be explained. After that, the authors should explain the effect of cleaning on improving the mechanical properties of the welded parts.

Reviewer 2 Report

The authors have substantially revised the text and tables in their manuscript. It is a pity that they did not find an opportunity to compare the method of cleaning metal surfaces using femto- and nanosecond pulses, on the one hand, and traditional physical and chemical technologies, on the other hand. The problem of ablation, which is typical for femtosecond pulses, was also left out of discussion. The term "ablation" occurs in the text only in the "key words" section. The absence of such a discussion can significantly reduce interest to this paper.

The article can be published in present form after correcting the data in the tables. The values of Peak power density in Table 5 and Average area of pores in Table 7 are given with an accuracy of approximately two orders of magnitude greater than the measurement accuracy.
